# "Are they trying to control us people?": News media coverage of COVID-19 lockdown tobacco sales ban in South Africa

Catherine O. Egbe[1,2]*, Senamile P. Ngobese[1,2], Hannah Barca[3], Eric Crosbie[3,4]

1 Alcohol, Tobacco and Other Drug Research Unit, South African Medical Research Council, Pretoria, South Africa, 2 Department of Public Health, Sefako Makgatho Health Sciences University, Pretoria, South Africa, 3 School of Public Health, University of Nevada, Reno, Nevada, United States of America, 4 Ozmen Institute for Global Studies, University of Nevada Reno, Reno, Nevada, United States of America

* catherine.egbe@mrc.ac.za

**Data Availability Statement:** All relevant data are within the paper and its Supporting information files.

## Abstract

The South African government imposed one of the strictest lockdowns in the world as part of measures to curb the spread of COVID-19 in the country, including a ban on the sale of tobacco products. This study explored news media coverage of arguments and activities in relation to the South African lockdown tobacco sales ban. We collected media articles published between 26 March to 17 August 2020, which corresponded to the period of the sales ban. Data were sourced via google search and snowball identification of relevant articles. Thematic analysis of data was conducted with the aid of NVivo. We analysed a total of 305 articles relevant to the South African tobacco sales ban during the lockdown. Six major themes were identified in the data: challenges associated with implementing the ban, litigation, and threats of litigation to remove the ban, governance process and politicization of the ban, pro and anti-tobacco sales ban activities and arguments and reactions to the announcement lifting the ban. The initial reason for placing the ban was due to the non-classification of tobacco products as an essential item. Early findings of a link between tobacco smoking and COVID-19 disease severity led to an extension of the ban to protect South Africa's fragile health system. Pro-sales ban arguments included the importance of protecting the health system from collapse due to rising COVID-19 hospitalization, benefit of cessation, and the need for non-smokers to be protected from exposure to secondhand smoke. Anti-sales ban arguments included the adverse effect of nicotine withdrawal symptoms on smokers, loss of jobs and the expansion of the illicit cigarette markets. Litigation against the ban's legality was a strategy used by the tobacco industry to mobilize the public against the ban while promoting their business through the distribution of branded masks and door-to-door delivery which goes against current tobacco regulations. The media could serve as a veritable tool to promote public health if engaged in productive ways to communicate and promote public health regulations to the general population. Engagement with the media should be enhanced as part of health promotion strategies.

**Funding:** COE and SPN's time were supported by the South African Medical Research Council. EC and HB's time was supported by the University of Nevada, Reno.

**Competing interests:** The authors have declared that no competing interests exist.

## Introduction

COVID-19 is an infectious respiratory disease spread through contaminated droplets [1]. Worldwide the healthcare burden of COVID-19 has been immense, with over 480 million confirmed cases, over 6 million deaths, and over 11 billion vaccine doses administered globally, as of March 2022 [2]. Increased hospitalization has been concerning for overwhelmed healthcare systems in countries strained by COVID-19 [3, 4]. While the relationship between tobacco use and COVID-19 infection remains debated, tobacco use has been identified as a risk factor for COVID-19 disease severity among those with the disease, including higher chances of hospitalization, ICU admissions, and death [3, 4].

In June 2020, the World Health Organization (WHO) issued a statement on tobacco use and COVID-19 affirming the link between tobacco smoking and COVID-19 disease severity and urged smokers to quit to decrease their chances of severe infection [5]. In response to these findings, the tobacco industry launched various misinformation campaigns, especially through social media, to influence public opinion on tobacco usage and its correlation to COVID-19 infection [6, 7] including promoting the idea that nicotine could be protective against COVID-19 infection. WHO advised researchers and the media to be cautious about spreading unsubstantiated information regarding COVID-19, tobacco, and nicotine, a likely warning against tobacco industry misinformation [5].

The African continent has the lowest prevalence of smoking compared to other regions, but the highest risk of increased smoking rates by 2025 [8]. In 2011, tobacco use prevalence in Africa was about 22% among those aged 15 years and older [9]. In 2019, smoking prevalence in sub-Saharan Africa was 17.5% among males and 2.94% among females 15 years and older [10]. Additionally, many countries in Africa still depend on tobacco, economically, which makes implementing tobacco control challenging [11]. The tobacco industry's interference on the African continent is rife with bribery, involvement in illicit trade and their use of the media as a propaganda tool, all in a bid to make government officials influence policies in their favour [12, 13].

Informed by early research on COVID-19, a few countries invoked various forms of tobacco bans as a mitigating effort to curb the spread of COVID-19 and reduce hospitalization and death from contracting the disease [6, 14]. Countries including Botswana, India, and South Africa implemented complete, but temporary, tobacco sales bans as part of their lockdown measures, in efforts to reduce the spread of the virus and protect their health system from collapse [4, 6, 7, 15, 16]. Other countries including Italy, Pakistan, Saudi Arabia, Spain, and the United Arab Emirates implemented some form of partial ban on tobacco products [4, 7]. In most cases the tobacco bans were implemented temporarily under essential goods restrictions or to curb an increasing healthcare burden [15, 17, 18].

However, other countries such as Kenya treated tobacco products as essential [3]. A few of these instances were due to tobacco industry interference. For example, in Italy, vape shops were declared essential after industry interaction with the government [18]. Concurrent interferences by the tobacco industry regarding COVID-19 restrictions were displayed in Argentina, Brazil, and Russia, many of which were conducted through media appeals and campaigns to gain public support [7].

South Africa has one of the highest tobacco use prevalence rates in Africa with about 37% of South African men and around 8% of women using tobacco [19]. As of February 2022, Africa had over 6 million cases and over 150,000 deaths from COVID-19, of which South Africa accounts for about 47% of the reported cases and about 59% of deaths documented [2]. South Africa also has a high prevalence of comorbidities that increase the risk of infection and death from COVID-19, including high prevalence of TB, HIV, obesity and diabetes

[19] predisposing the country to a heavy COVID-19 burden. This may partly explain the reason the country has the majority of reported COVID-19 cases on the continent, though, this may also be indicative of South Africa having the highest rates of testing in the continent [20].

Following the first case of COVID-19 reported in South Africa in March 2020, the government quickly imposed some of the strictest lockdowns in the world in order to curb the spread of the virus [14]. South Africa's Disaster Management Act (DMA) was enacted in March 2020, which included a 5-tier lockdown level (5 being the strictest) [17]. Objectives of the various tiers of lockdown are shown in Fig 1.

Because tobacco and vape products did not fall under the definition of essential goods, defined as food, cleaning and hygiene products, medical, fuel and basic goods (including electricity and airtime)" a temporary ban was implemented on the sales of tobacco and vape products from 26 March and this lasted until 17 August 2020 [17]. Alcohol was also banned during the lockdown while nicotine replacement therapy, which is not legally considered a "tobacco or vape product" was excluded as they would fall under the 'medicines' category of essential goods. As lockdown continued and was downgraded to tiers 4 and 3, the South African government kept the tobacco ban in place following guidance from WHO and South Africa's own medical researchers [5, 21]. Interference from the tobacco industry through media campaigns, litigation and threats of litigation against the tobacco ban and subsequent public backlash against the ban created tension in South Africa, and the media played a large role in how the public perceived the ban.

To the best of the authors' knowledge, this will be the second known article to examine media coverage on tobacco bans during the pandemic lockdown, the first pertaining to India's alcohol and tobacco bans [22]. This study investigates news media coverage of arguments and activities in relation to the South African lockdown tobacco sales ban from March to August 2020.

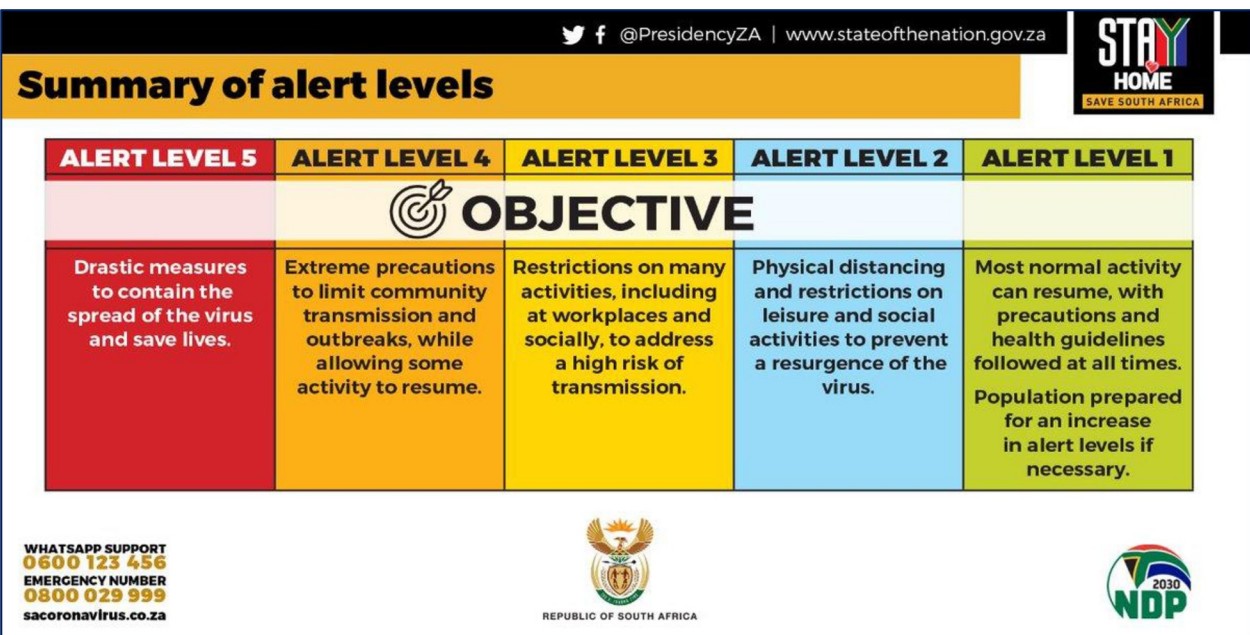

**Fig 1. Summary of alert levels during South Africa's COVID-19 lockdown (Source: Official twitter page of the presidency of South Africa).**

## Methods

Between June and September 2020, we collected news articles published between 26 March 2020 and 17 August 2020 (the time period the ban was in effect) [23] using Google search engine. The w3newspaper website, which is a directory of world newspapers, magazines, news sites, and publishers, reports that there are 69 South African news outlets online, including both English and other South Africa languages [24]. While other studies have selected newspapers based on website traffic [25], for this study the search was not restricted to popular newspapers since a number of regional newspapers are popular in their local regions and were considered important for this study. We considered all media outlets with online presence (print, radio and TV) except social media (Facebook, Twitter, Instagram etc) which would have required a different methodology to be included. The search terms used were 'Tobacco and COVID-19 lockdown', 'Smoking and COVID-19 lockdown', 'Cigarettes and COVID-19 lockdown', 'Vaping and COVID-19 lockdown', 'Cigarette ban lifted' AND 'South Africa'. The exclusion criteria were articles which did not focus on the tobacco sales ban in South Africa, articles published outside of South Africa and those not in English. There were 305 articles that met the inclusion criteria (see Fig 2) and were included in the analysis.

Relevant information was extracted from all 305 articles into a spreadsheet, including: date of publication, name of author, name of media outlet, title of publication, section of news articles published and URL (S1 Table).

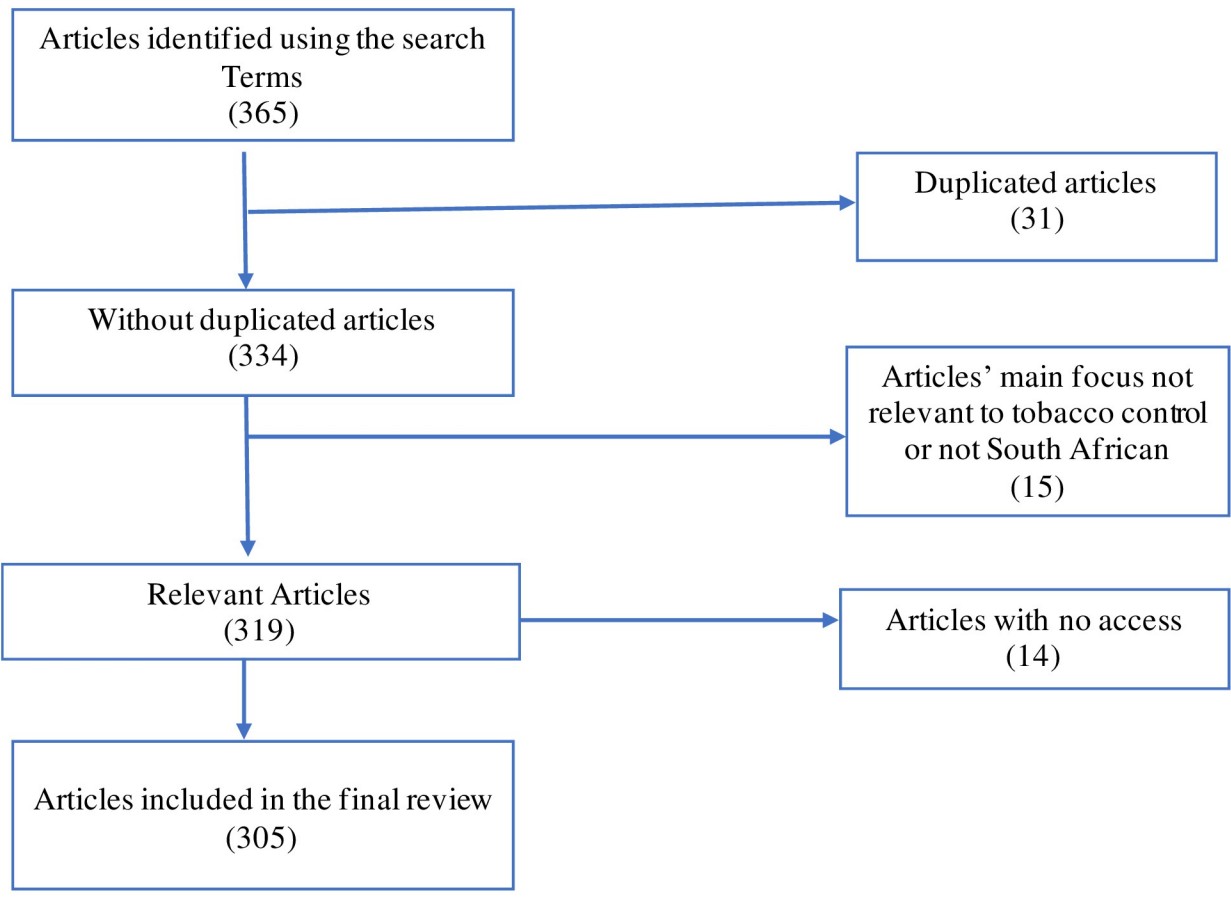

**Fig 2. Flowchart of news articles selected for analysis.**

The news media entity publishing the most articles on the sales ban was Times Live (18.4%, n = 56), followed by The South African (12.1%, n = 37) and IOL (9.5%, n = 29) (Fig 3a). Though the ban was imposed in the last week of March 2020, 69% (n = 209/305) of the articles were published between June and August (Fig 3b).

The 305 articles were saved directly from their websites and exported to NVIVO for analysis. Data were qualitatively analysed using thematic analysis based on the procedure described by Nowell et al. [26, 32]. This type of analysis is suitable for this study due to its usefulness in investigating varying perspectives and providing a rich and detailed account of the data [26]. Multiple coders were used to extract themes and verify classification of themes.

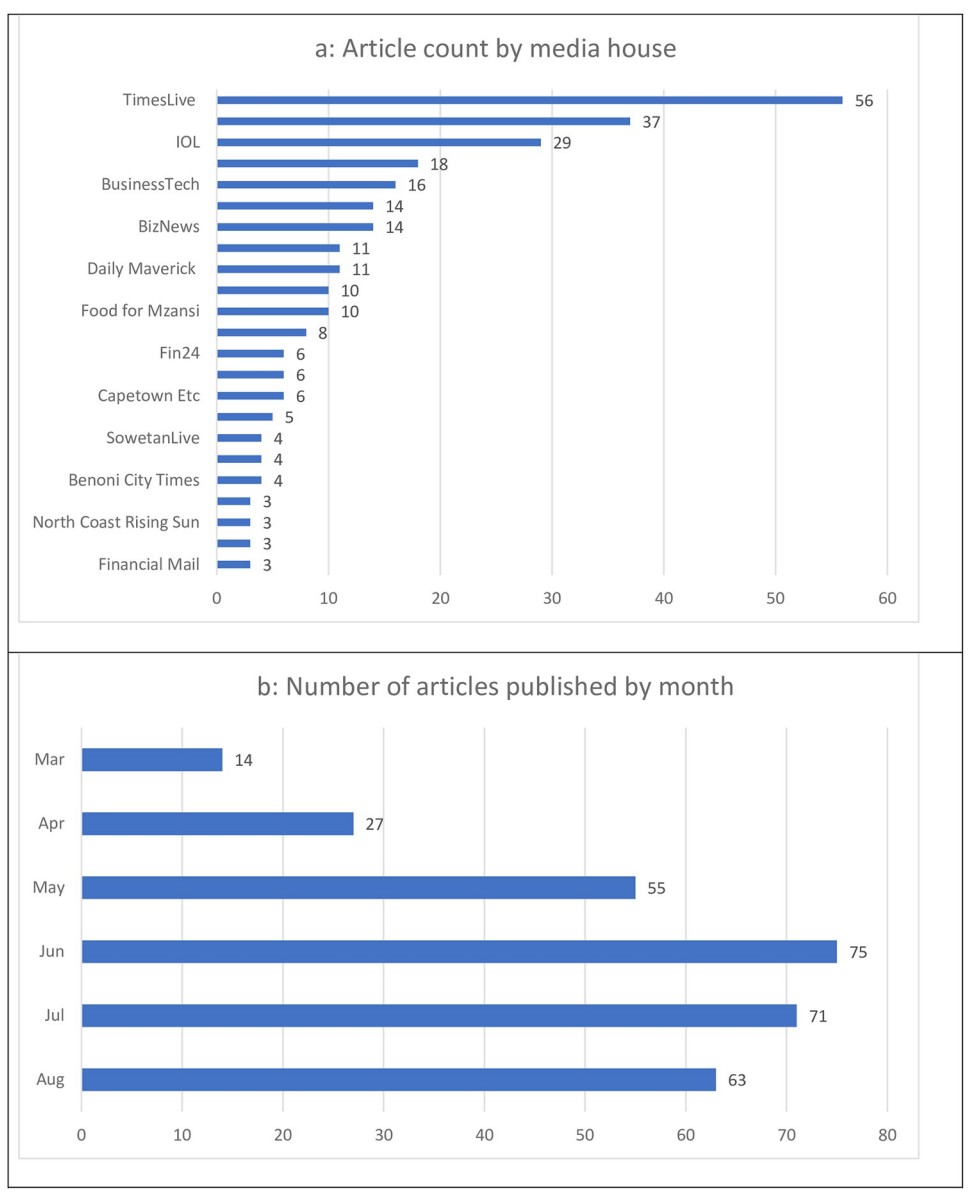

**Fig 3. Distribution of news media publications by month of publication and media house.** a. Publication count by media house; only newspapers with more than two published articles are included in this graph, b. Publication count by month during the period of the tobacco sales ban.

## Results

The timeline of activities during the period of the tobacco sales ban in South Africa is presented in Table 1.

Six major themes were derived from the qualitative analysis of the news media publications. These themes are, challenges associated with implementing the ban, litigation and threats of litigation to remove the ban, governance processes and politicization of the ban, pro-tobacco

**Table 1. South African COVID-19 Tobacco Sales Ban Timeline (5 March 2020–17 August 2020).**

| Date | Event(s) |
|---|---|
| 5 March 2020 | First COVID-19 case in South Africa confirmed by Health Minister Zweli Mkhize |
| 15 March 2020 | President Cyril Ramaphosa declares National State of Disaster as well as travel bans/restrictions and school closures to begin 18 March |
| 17 March 2020 | First meeting of the National Command Council on COVID-19 |
| 23 March 2020 | President Ramaphosa announces a 21-day lockdown to start at midnight on 26 March |
| 25 March 2020 | Trade and Industry Minister Ebrahim Patel makes a statement about upcoming regulations and essential services |
| | Tobacco and vaping products considered non-essential products under Level 5 lockdown |
| 27 March 2020 | Level 5 lockdown begins |
| | Health Minister Mkhize states that smokers are a part of high risk/vulnerable population for COVID-19 |
| | First confirmed death from COVID-19 in South Africa |
| 30 March 2020 | Garden Route reported to be selling tobacco products |
| | Ban reversal rumors dismissed by Police Minister Bheki Cele |
| 1 April 2020 | Western Cape allows purchase of cigarettes alongside other essential items |
| 2 April 2020 | Police Minister Cele announces that national regulations override provincial ones, thus the Western Cape is breaking lockdown rules |
| 23 April 2020 | President Rampaphosa announces lift of tobacco ban to begin 1 May as regulations are lowered from Level 5 to Level 4 |
| 27 April 2020 | Co-operative Governance minister Dlamini Zuma announces reversal of President's April 23rd statement (ban will remain in place during Level 4 lockdown) |
| 30 April 2020 | BATSA sends a formal letter to the government and threatens legal action |
| 4 May 2020 | FITA files for legal action against ban for reasonableness |
| 6 May 2020 | Co-operative Governance minister, Dr Dlamini-Zuma files responding documents to court on behalf of the government to oppose FITA |
| 6 May 2020 | BATSA withdraws bid to file legal action against the government |
| 1 June 2020 | Lockdown lowered from Level 4 to Level 3, ban remains in place |
| 2 June 2020 | State attorney requests postponement of the FITA court case from 9/10 June to 22 June |
| 3 June 2020 | BATSA files legal action against ban for unconstitutionality |
| 10 June 2020 | Oral arguments heard for FITA case |
| 26 June 2020 | Court rules in government's favor on tobacco ban against FITA, for reasonableness |
| | Court approves delay of BATSA case from 30 June to 5/6 August 2020 |
| 5 August 2020 | Oral arguments heard for BATSA case |
| 15 August 2020 | President Cyril Ramaphosa announces that SA would move to level 2 lockdown at midnight 17 August 2020 and the sale of tobacco products would resume |
| 17 August 2020 | Ban is lifted as restrictions are lowered from Level 3 to Level 2 per President Ramaphosa's announcement |

FITA–Fair-trade Independent Tobacco Association; BATSA–British American Tobacco South Africa

**Table 2. Summary of themes and subthemes.**

| S/No | Themes | Subthemes |
|---|---|---|
| 1 | Challenges associated with implementing the ban | 1. Controversy around the classification of tobacco products as non-essential products<br>2. Challenges interpreting the Disaster Management Act<br>3. Compliance with the ban<br>4. Tobacco industry tactics during the ban |
| 2 | Litigation and threat of litigation to remove the ban | 1. Litigation and threats of litigation<br>2. Reactions to court processes against Ban<br>3. Arguments for and against the ban during litigation |
| 3 | Governance processes and politicization of the ban | 1. Conditions for lifting the ban<br>2. Criticism of government actions during the ban<br>3. Political polarization of support and opposition to the ban |
| 4 | Pro-tobacco sales ban arguments and activities | 1. Health arguments in support of the ban<br>2. Economic and illicit trade arguments in support for the ban<br>3. Correcting misinformation about the ban |
| 5 | Anti- tobacco sales ban activities and arguments | 1. Protesting the ban<br>2. Social and health arguments against the ban<br>3. Human rights and harm reduction arguments against the ban<br>4. Economic and illicit trade arguments against the ban |
| 6 | Reactions to the announcement lifting the ban | 1. Tobacco Industry accepting the lifting of ban with caution<br>2. Continuation of litigation post tobacco ban |

sales ban arguments and activities, anti- tobacco sales ban activities and arguments, and reactions to the announcement lifting the ban. A summary of the themes and subthemes is presented in Table 2.

## Challenges associated with implementing the ban

**Controversy around the classification of tobacco products as non-essential products.** The government banned the sale of tobacco products because they were not categorized as essential items in the Disaster Management Act (DMA) 2020 which implemented the lockdown [17]. The reason for this non-classification was because tobacco products do not *"by their nature, fall into the same category as goods which are life-sustaining or necessary for basic functionality"* (Article 162). An article published in the Daily Maverick reported that, *"a prohibition on the sale of cigarettes was not spelt out in the regulations. Instead, ministers said cigarettes were not considered "essential" and so could not be sold."* (Article 31) The announcement was followed by a public debate whether or not tobacco products should have been regarded as essential goods. An article by 2Oceansvibe summarized the controversy on tobacco and other issues related to the lockdown [27]. This publication reported on how the Western Cape Government (one of nine South African provinces) gave contradicting explanation to the DMA including lifting of the ban against the sale of tobacco products on 1 April 2020 as well as the response of the Police Minister confirming that the Western Cape government could not implement provincial regulations which differ from the national government [27].

**Challenges interpreting the Disaster Management Act (DMA) 2020 and its regulation.** There was confusion concerning the interpretation of the regulation guiding the DMA. For example, the Premier of the Western Cape said *"[t]he agreement was that as long as it*

*[tobacco products bought] is with shopping items, then it would be fine. It is not about just going and buying cigarettes. It's not about the opening of tobacconists or cigarette-specific shops. It is part of your buying essential items'"* (Article 23). There were also media reports suggesting that the ban had been lifted by the provincial government in the Western Cape, *"The Western Cape government has lifted the ban on sale of cigarettes in major essential goods stores during lockdown."* (Article 24). This interpretation of the regulations to the DMA contradicted the interpretation by the national government as mentioned earlier.

The varying interpretation of the ban in some provinces led to challenges with implementing the ban. Despite these challenges the police minister publicly corrected the interpretation by the Western Cape Premier. The police minister was reported as saying that *"the ban on the sale of cigarettes has not been lifted, as it [cigarette] is not an essential item"* (Article 7) and the minister urged *"businesses in the* [Western Cape] *province 'not to listen to people who tell them wrong things"* (Article 127).

**Compliance with the ban.**   At the beginning of the lockdown, it was reported that the public had responded positively by adhering to the lockdown restrictions and showed support for the ban. An article reported that *"Most South Africans are adhering to lockdown rules when it comes to tobacco products that have been banned. A survey by the Human Sciences Research Council (HSRC) has found 88 percent of people were not able to buy cigarettes."* (Article 60). There were also reports that tobacco was more accessible to some categories of people. A news report mentioned that *"Cigarette buying was more prevalent among those who were able to buy alcohol than those who were not able to buy alcohol -72 percent of people who bought alcohol also bought cigarettes."* (Article 60). Alcohol was also banned for most of the time tobacco products were banned. However, the media also reported that in some places (e.g., Western Cape) the ban was being flouted with people buying tobacco with groceries (likely due to the Premier's announcement about the ban being lifted), while in other places (e.g., Johannesburg) tobacco was sold in spaza (tuck) shops.

**Tobacco industry tactics during the ban.**   Despite the regulations put in place by government to stop the sale (and manufacturing) of tobacco products, the tobacco industry was reportedly producing and marketing their products during this time. During this period, an article reported that British American Tobacco South Africa (BATSA) *"has not appeared to have stopped producing cigarettes during the ban, with SARS [South African Revenue Service] reportedly raiding a BAT factory which continued to produce cigarettes during the ban"* (Article 192). News media also mentioned strategies used by the industry under the guise of corporate social responsibility to launder their image and promote their business during the ban. On 30 May, 2020, TimesLive reported that *"As part of their cunning and deception strategy disguised as a social responsibility, the industry has offered free branded masks and delivery to your door during quarantine"* (Article 83). The industry also tried to market electronic cigarettes (though banned) as an alternative to cigarettes as well as institute court cases against the government (see next section).

## Litigation and threat of litigation to remove the ban

**Litigation and threats of litigation.**   Following the announcement of the ban, BATSA and the Fair-Trade Independent Tobacco Association (FITA), a trade organization of local tobacco manufacturers [28], threatened to sue the government arguing that the ban was unconstitutional and tobacco products should be categorized as essential goods. However, only FITA carried out its initial threat to institute legal action against the ban in May 2020, BATSA withdrew its bid to sue the government at first. In an article published by IOL, it was mentioned that that *"the case is being brought against government by the Fair-Trade Independent Tobacco*

*Association (FITA) which is arguing that the banning of the sale of cigarettes is irrational"* (Article 117). The government won this lawsuit and FITA appealed the judgement. In their appeal, FITA accused the court of making errors of interpretation when reading the Disaster Management Act (Article 201). The government again won the appeal, and the ban was upheld. An eNCA article reported that *"Fita approached the court after the high court dismissed its legal challenge on the current tobacco ban. According to the North Gauteng High Court, the association failed to show why an appeal should be heard"* (Article 214). FITA appealed to the Constitutional court afterwards.

After the ban was extended when the lockdown was downgraded to level 3 in June 2020 (which allowed more business activities in most sectors of the economy), other tobacco interest groups led by BATSA, Japan Tobacco International (JTI), and tobacco farmers amongst others, also sued the government to overturn the ban. The media, quoting the litigants, explained that part of the reason for the legal battle was because the industry received no response from government upon request for an explanation why the ban was being maintained even after other previously banned products were now being sold. An article published by IOL quoted Johnny Moloto, the spokesperson of BATSA, thus *"Given the situation, and the lack of any response from the government, despite our ongoing efforts to engage with them, we are now commencing urgent legal proceedings"* (Article 97).

**Reactions to court processes against ban.** There was a wide coverage of both the FITA and BATSA court cases in the media during this period including arguments for and against the parties in the court cases (government and the tobacco industry). Some of the arguments included accusations that the court had been captured (in reference to the state capture phenomenon currently under investigation in South Africa [29]) due to delays or postponements of court hearings. A news article reported that "*The giant tobacco producer and the government are pointing fingers at each other over the delay of another case that will determine whether or not the ban on the sale of tobacco products should continue*" (Article 167). There were varying opinions as to who is to blame for the delays in the court cases. An article by Business Day reported that *"The government has made a second attempt to delay the hearing of the first major legal challenge to the controversial cigarette ban, appealing to court again for the matter to be delayed"* (Article 102). BATSA condemned the delay through their spokesperson Johnny Moloto saying, *"This delaying of justice and a resolution of this issue is inexplicable"'* (Article 150). BATSA also wrote a letter protesting the postponement of the case (Article 150). On the other hand, the government had accused BATSA of having caused the postponement in court by introducing new evidence and not warning them in time. A published statement from Minister Nkosazana Dlamini-Zuma (Minister of Corporative governance and traditional affairs (COGTA), in charge of the DMA) said *". . .the complainant has not forewarned the state about new evidence, contained in replying papers delivered on Wednesday afternoon"* (Article 167).

**Arguments for and against the ban during litigation.** Reasons reported as part of government's consideration before placing the ban include health and economic factors. In the judgement delivered on the FITA vs South African government case, TimesLive reported that the court found the government took into consideration, *". . .relevant factors such as the economic impacts of the prohibition, the potential health and psychological impact on smokers as well as the illicit trade in cigarettes* (Article 159).

Several news outlets reported that the industry questioned the scientific evidence linking smoking and COVID-19 disease severity as presented by the government in court. A media article reported the industry as saying that, *"Some studies do not differentiate between current and past smokers, none of the datasets stratify smokers according to pack-year-history (a measure of quantity and duration of smoking)"* (Article 121). Another article published by The South African reported that the argument made by FITA was that *"The surveys relied upon by*

*the minister were themselves without probative value and particularly unscientific and crude"* (Article 276). FITA argued that *". . .the World Health Organisation (WHO) stated last month that no peer-reviewed studies had been published to ascertain these facts. . . this meant there were no conclusive studies to rely upon showing smokers were more at risk [of COVID-19 disease severity]"* (Article 124). In their report of court proceedings, Fin24, quoted Advocate Arnold Subel SC, representing FITA saying, *". . .more peer-reviewed studies would have to be done to support the view that smokers are more likely to contract severe cases of Covid-19, as the current evidence is inconclusive"* (Article 163). Fin24 also quoted the advocate representing BATSA (Advocate Alfred Cockrell SC) thus *"Cockrell called into question government's medical basis for the ban, saying that, while there was no question that smoking is harmful to health, the core question is whether there is an association between smoking and the contraction of a more severe form of Covid-19. According to Cockrell, there is a medical dispute regarding this"* (Article 277).

The industry described the scientific evidence linking smoking with COVID-19 disease severity as incorrect, sloppy and outdated. A psychiatrist named Mike West, made this claim in his submission to the court on behalf of FITA in response to the government's supplementary affidavit. The media reported that Mike West's *analysis exposes how the government's arguments include flawed estimates of how many SA [South African] smokers will get Covid-19, incorrect study references, false claims, outdated science from the 1990s, and studies cherry-picked to show the negative effects of smoking on Covid-19"* (Article 121).

A potential protective nature of nicotine against COVID-19 as reported in some pre-print academic papers was also presented in court as reported by some media outlets. TimesLive reported that '*The tobacco industry has countered this [the ban] by pointing to other peer-reviewed studies that suggest smokers are less likely to be hospitalised when testing positive for Covid-19 and that nicotine may offer potential protection against the disease'* (Article 173). Another argument was that the damage caused by smoking takes years to accumulate therefore it cannot be reversed overnight, making the ban not a good idea. News24 cited a scientist who said that *". . .but as to whether stopping smoking would protect you against Covid-19, the answer was 'no'—because the damage caused by smoking does not occur overnight, according to Madhi."* (Article 180).

### Governance processes and politicization of the ban

**Conditions for lifting the ban.** During the period the ban was in place, there were reports speculating about what should be considered before the lifting of the ban. The health risk, the impact on the economy and the progression of the COVID-19 pandemic in South Africa were some of the factors proposed to be examine before lifting the ban. The conditions spelt out by the South African president as reported by the media include *". . .the progression of the disease in South Africa, the readiness of our health systems and evolving knowledge on the nature and impact of the virus itself"* (Article 117). There were also reports about the uncertainty of when the ban would be lifted and who would make the decision on when the ban is to be lifted. An article by Eyewitness News reported that *"According to government, the bans on these products will remain in place throughout the lockdown period and at this stage, with over 380,000 positives recorded, there's no telling when lockdown will be declared over"* (Article 234).

**Criticism of government actions during the ban.** There were many reports critical of the government's decision to impose the ban. The government was accused of not making a "good decision" by banning the sales of tobacco products. The government was also accused of not being able to give satisfactory reasons for imposing the ban as well as cherry-picking scientific evidence to support the ban. An important development during this period was the announcement that the ban would be lifted on 1 May 2022 as the country moved from lockdown level 4

to level 3. However, this was subsequently reversed leading to amplified criticism of the government and accusations that the government was controlling people. A news article published on the 30 April 2020 reported a community member saying *"My mother didn't even tell me I can't smoke, Ramaphosa said we can start buying cigarettes from the 1st of May and now they change their minds. Are they trying to control us people?"* (Article 41).

**Political polarization of support and opposition to the ban.**    The media also reported the ban as a political battle between the various political parties with political party members tending to align themselves with certain positions about the ban. An article in the Daily Maverick reported that *"[t]here is also a strong party-political dimension, with ANC supporters being far less likely (15%) to want the ban to be lifted, than those who said they would vote EFF (25%), DA (62%) or 'another party' (42%) if an election were to be held"* (Article 246).

## Pro-tobacco sales ban arguments and activities

**Health arguments in support of the ban.**    There were varied arguments put forward in support of the ban in the media, including health reasons why the ban was needed. A report by Fin24 stated that *"The state has defended the ban as necessary for health reasons"* (Article 201). Quoting from the responding affidavit submitted by the COGTA Minister, Dr Dlamini-Zuma, TimesLive reported the following *"In a situation of evolving scientific knowledge and with infection numbers rising, a responsible government has to take a cautious approach. Prohibiting the sale of tobacco products during lockdown serves to reduce these risks, not only in respect of smokers themselves, but also those who would otherwise be exposed to second-hand smoke under lockdown conditions"* (Article 108). The severity of COVID-19 among smokers and how the ban can assist smokers to quit were highlighted in several media reports. The South African reported that *"In defense of government's position, Dlamini-Zuma [COGTA Minister] has argued that the ban has afforded many South Africans an opportunity to quit smoking and has, thereby, reduced the burden on the healthcare system"* (Article 156).

**Economic and illicit trade arguments in support for the ban.**    Interestingly, a decrease in illicit trade was put forward to argue in support of the ban. It was reported that a decrease in smoking prevalence would lead to a decrease in illicit trade. An article reported that *"If fewer South Africans smoke, then the consumer demand for illicit cigarettes will fall and this will be accompanied by a decline in the 'illicit trade'"* (Article 91).

**Correcting misinformation about the ban.**    Opposition to the ban also stemmed from the misinformation that the ban was a prelude to total prohibition of tobacco use in the country as was seen in the United States with the prohibition of alcohol in the 1920s and early 1930s. Kopo Mapila (a former public sector policy analyst) writing in the Daily Maverick discussed the dissimilarity between these two scenarios, *"... the strongest argument against banning cigarettes is built on the US experience of Prohibition.... While South Africa has a well-established illicit cigarette trade and certainly its own economic woes, the situation is hardly comparable when looking at the costs against their respective time scales. Prohibition in the US lasted for 13 years. The cigarette ban, on the other hand, is an emergency measure that will last, at the extreme, around 18–24 months–the estimated vaccine development period. Unlike Prohibition, the ban is not implemented as an indefinite regulation to manage 'immoral behaviour; it is, rather, a short-term measure to reduce the number of people requiring hospitalisation and stretching capacity"* (Article 126).

## Anti- tobacco sales ban activities and arguments

**Protesting the ban.**    Apart from legal actions against the ban led by various tobacco industry players, petitions and protest were also among the anti-tobacco sales ban activities which

took place during the period the ban was in place. There was a *Dear Mr. President* protest held to get the ban lifted and people were mobilized to show support for the protest on social media. A newspaper article reported that *"People who can't make it are being asked to upload selfies with the hashtag #SmokersProtest, to ensure a digital presence is felt."* (Article 117).

**Social and health arguments against the ban.** Anti-tobacco sales ban argument centred around the social, health and economic issues which were claimed to result from the ban. The ban was reported as a social issue needing to be addressed because it caused tensions in households (due to nicotine withdrawals) and negative activities (smokers turning to the illicit cigarette market) and reveals the inequality in the society. An article by Kopo Mapila published on 10 June 2020, describes two groups of opposition voices to the ban. The *". . .My Lungs, My Choice Movement, and the second, the 'Black Market Brigade", whose concerns rest on corruption and criminality in the black market"*, he therefore suggested that *". . .a moral approach to public policy in South Africa should give short shrift to both sets of complaints"* (Article 126). The South Africa Drug Initiative called for the ban to be lifted because *"It believed it discriminated against the poor, puts the health of the mentally ill and marginalised at risk, and gifts crime syndicates and gangs another source of income"* (Article 22). It was also reported that the ban encourages movement (in search of cigarettes) during the lockdown leading to the spread of COVID-19, and that it increases the sharing of cigarettes (due to its scarcity), putting the health of the people at a greater risk which is not beneficial to the health system.

**Human rights and harm reduction arguments against the ban.** The ban was said to infringe on the rights to sell tobacco, which affected tobacco farmers and sellers. BATSA stated in their court papers that they *". . .challenge the constitutionality of the ban as an 'irrational interference' on the rights of tobacco farmers, manufacturers, wholesalers and retailers, and an infringement on the dignity, privacy and physical integrity of smokers"* (Article 98).

Electronic cigarette industry allies also presented electronic cigarettes as a safer alternative to traditional tobacco products hence should have been excluded from the ban. An article by The South African reported that the African Harm Reduction Alliance implored the government to *". . .consider how vapers can be assured continued access to these less-harmful products during this critical time"* (Article 01).

**Economic and illicit trade arguments against the ban.** Many arguments were made to indicate that the ban on the sale of tobacco products had a negative impact on the economy due to an increase in illicit trade and loss in tax revenue. According to an article by Business Day *"The ban, which has resulted in a spike in the illicit trade of cigarettes, has also cost the SA Revenue Service billions of rand in lost excise duty."* (Article 115).

Argument about job losses in the farms was also made in opposition to the ban. TimesLives (Sunday Times) reported that *". . .the tobacco sales ban had left farmers and farm workers in despair. 'It has destroyed marketing opportunities, income streams, jobs and livelihoods in the primary production sector'"* (Article 158).

## Reactions to the lifting of the ban

After almost five months, the South African president announced on 15 August 2020 that the ban would be lifted by midnight of 17 August 2020. A media outlet, Cape(town)etc, reported that *"During his national address on Saturday evening, President Cyril Ramaphosa moved the country to Level 2 restrictions from midnight on Monday, August 17. Further to this he announced that restrictions on tobacco products will be lifted, and the sale of alcohol will be permitted from Monday to Thursday"* (Article 294). The news was received with cautious excitement by the tobacco industry, but they indicated they would continue the court case against

the government. An article reported that *"The tobacco industry has cautiously welcomed the announcement that cigarettes can be sold from Tuesday, but the ban on trade for five months has left it reeling"* (Article 295). FITA subsequently announced that they would continue with their appeal of the case brought against the government at the Constitutional court. An article reported that *"The Fair-Trade Independent Tobacco Association (FITA) has announced that it will continue with its legal action, where it's appealing against the High Court decision to dismiss its court bid to have the cigarette sales ban overturned"* (Article 303). This case was later settled out of court [30].

## Discussion

We qualitatively explored the discussions and activities in the media space during the period the COVID-19 lockdown tobacco ban was in place in South Africa as a way to appraise the issues of interest to the public during this period. Results show that the sales ban of tobacco products, during the COVID-19 lockdown in South Africa, was a topic vastly covered by the media in the country and peaked in the month of June when two court cases against the ban were being heard. According to media reports, the ban brought about controversy regarding the classification of tobacco products whether or not it is essential. There were challenges associated with implementing the ban, where in some places the ban was being flouted due to provincial governments controversial lifting of the national ban. The media covered the court cases against the ban, reported on the delays of the court hearings, accusations about the court being 'captured' (referring to the ongoing 'state capture' investigation in South Africa), the industry introducing more evidence at the last minute to delay the case, and evidence submitted by government in support of the ban and government winning the FITA case. Anti-tobacco sales ban activities and arguments together with pro-tobacco sales ban activities and arguments were major themes also covered by the media.

The tobacco sales ban implemented in South Africa between March and August of 2020 was received with both support and opposition by the public. It started with the non-classification of tobacco as an essential good, hence halting sales of tobacco and vape products and continued when early evidence suggested a link between tobacco use and COVID-19 disease severity [5]. The government had argued that the ban would lower the burden on the health system [14], however, there were some implementation challenges [31, 32] especially in the Western Cape, which is also the province with the highest rates of tobacco use in South Africa [19].

The results show how strategic the industry was in creating a narrative that the public should follow, including organizing online protests and promoting the notion that the ban has led to an increase in illicit trade and loss of jobs and tax revenue [33]. While the tactics used by the tobacco industry in South Africa are common strategies used around the world, especially in low and middle-income countries (LMICs) [34], the anti-sales ban theme (Table 2) carries examples of the industry's discursive strategies in the media [35]. Research has shown that when faced with policies regulating their products, the industry uses exaggeration of potential cost of the policy, at the same time denying its potential benefits [36]. A new bill, (the Control of tobacco and electronic delivery systems bill) which would comprehensively regulate tobacco and electronic cigarettes is presently being processed in South Africa. The tobacco industry had also tried to make the pubic believe that the bill will cause severe harm to the economy, society, and public health [36]. This bill, if passed into law, would enable South Africa to regulate electronic cigarettes, ban sales via vending machines, remove designated smoking areas thereby enforcing 100% smoke-free public places and introduce standardize packaging among other provisions [37].

FITA and BATSA were heavily involved in trying to repeal the tobacco ban and increase public disapproval. One of the strategies used by the tobacco industry was the spreading of misinformation about the South Africa tobacco ban being a lead-up to a prohibition of tobacco use in the country through to permanent legislation. As seen in other LMICs, this strategy was designed to incite fear that the public's rights and autonomy is being infringed upon [38, 39]. LMICs, which include South Africa, are usually targeted by the tobacco industry due to the rapid increase of the youth population, widespread poverty, and poor legislation [40, 41].

Media coverage and campaigns can be said to be influential in how the public perceived the ban during South Africa's COVID-19 lockdown protocols. Previous research suggests that one of the tobacco industry's tactics is to promote a positive coverage of their activities and brand in the media [35]. The industry uses its resources to be more visible in the media in order to silence any counter voices. Our results show that health care professionals, researchers, and public health experts were also visible in the media. These health experts tried to educate the public on why the temporary ban was put in place and how it could improve the COVID-19 burden, though they may have had lower coverage by the media. The ban was supported by health experts for its usefulness in curbing the spread of COVID-19 from both a biological standpoint of reducing smoking which is a risk factor for COVID-19 disease severity, and behaviourally for reducing the movement of persons thus reducing exposure to the coronavirus.

Gilmore and colleagues discussed tobacco industry strategies being utilized to prevent public health improvements in the tobacco sector, many of which were employed during the South African lockdown ban [41]. For example, the tobacco industry has been an active partner in harm reduction campaigns; alternative tobacco and nicotine product creation; and engaging in deceptive government interaction [41]. These were all present in South Africa in the forms of misinformation campaigns via petitions and social media, protests promoting industry-funded research, advocating that electronic cigarettes should have been excluded from the ban as a harm reduction tool, issuing threats of litigation and active litigation against the government during the ban. While there are arguments for and against the use of electronic cigarettes as a harm reduction tool among scientists, the arguments in the South African media did not go deeper than the mention of a call to the government to exclude electronic cigarettes from the lockdown tobacco ban. Negative media coverage may have played a role in public disapproval of the ban. Such disapproval was lower at the onset of the ban but grew as it was extended [42]. The evolving nature of social media, which is known to be used by the industry to promote their products, was used to fight against the ban as the industry knew they could reach millions of people daily [35]. In our results, petitions and protests using social media were used to try to get the ban lifted. The tobacco industry uses media to manipulate public opinion about tobacco control and to garner the support of people who oppose government's "intrusion" in their business [35]. Media as a tool for interference and promotion may be an important tool for the future of public health as it has become a viable tool both for the promotion and discounting of public health [43, 44].

Despite the opposition to the ban, the move by the South African government to protect their citizens received some global recognition and commendations [14]. The temporary sales ban showed that the government is capable of putting the country's health first regardless of industry's pressure to do otherwise. It is hoped that this political will would be extended to the Control for Tobacco Products and Electronic Delivery Systems (CTPEDS) bill which is currently facing fierce opposition from the industry. The CTPEDS bill seeks to remove provisions for designated smoking areas to implement 100% smoke-free public places, stop the sale of tobacco products via vending machines, and implement standardized packaging (amongst other provisions) [37]. Thus, the act of having a ban in place, enforcing it regardless of the

pushback, reveals that the South African government can indeed enforce stricter laws on tobacco use.

## Strengths and limitations

This study utilized code verification to ensure the six themes used were not biased to one researcher. The number of articles included in this study is 305 and their publication spanned the entire period when the ban was in place. The discussion about the ban has continued till date though our study cut-off point was at the date the ban was lifted, however the period captured in this paper was the most crucial for understanding the pulse of the public and industry activities in relation to the ban. Social media attracts huge readership in South Africa; however, most media outlets also maintain social media handles through which they further disseminate their news publications. Social media was not included in this study due to the difference in methodology and to avoid duplicating publications. Given the increasing usage of social media in South Africa and globally, for future research and practice it is recommended to consider a distinct study on social media coverage and reactions to the COVID-19 lockdown tobacco ban and compare such results to our findings. Though a quantitative investigation and weighting of the media articles could have provided more depth to this analysis, this was a challenge to conduct because most articles covered varied aspects (both positive and negative) of the pulse of the public in relation to the ban.

## Conclusion

This study reveals the trends and themes of newspaper media coverage about the temporary ban on the sale of tobacco products in South Africa during lockdown. It shows the narrative the tobacco industry managed to push using the media and how they fought against the ban. At the same time, it also shows the activities and arguments of tobacco control advocates, and the government during the period of the tobacco ban. The media, being an important component of society, needs to be included in tobacco control efforts. Media platforms can be used to send the right information about government policies and programmes that would benefit tobacco control and public health in general. The media can play a huge role in spreading awareness or spreading misinformation as well as mobilizing the public in support for or against public health policies. The media may have been used to mobilize citizens against the ban. The health benefits of smoking cessation were scarcely discussed in the media during the period suggesting that this is a missed opportunity for public health. Industry monitoring and stricter laws which would prevent the tobacco industry from using the media to its advantage are needed while engagement with the media should be enhanced as part of health promotion strategies.

## Supporting information

**S1 Table.**
(XLSX)

## Author Contributions

**Conceptualization:** Catherine O. Egbe.

**Data curation:** Catherine O. Egbe, Senamile P. Ngobese.

**Formal analysis:** Senamile P. Ngobese, Hannah Barca.

**Methodology:** Catherine O. Egbe.

**Validation:** Catherine O. Egbe, Hannah Barca.

**Writing – original draft:** Catherine O. Egbe, Senamile P. Ngobese, Hannah Barca, Eric Crosbie.

**Writing – review & editing:** Catherine O. Egbe, Senamile P. Ngobese, Hannah Barca, Eric Crosbie.

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
