## [Decision Letter · Decision Letter 0]

17 May 2022

PONE-D-22-09801“Are they trying to control us people?”: News media coverage of COVID-19 lockdown tobacco sales ban in South AfricaPLOS ONE

Dear Dr. Egbe,

Thank you for submitting your manuscript to PLOS ONE. After careful consideration, we feel that it has merit but does not fully meet PLOS ONE’s publication criteria as it currently stands. Therefore, we invite you to submit a revised version of the manuscript that addresses the points raised during the review process.

We look forward to receiving your revised manuscript.

Kind regards,

Michael Cummings, PhD

Academic Editor

PLOS ONE

Journal Requirements:

"COE and SPN’s time were supported by the South African Medical Research Council. EC and HB’s time was supported by the University of Nevada, Reno."

"COE and SPN’s time were supported by the South African Medical Research Council. EC and HB’s time was supported by the University of Nevada, Reno."

Reviewers' comments:

Reviewer's Responses to Questions

**Comments to the Author**

1. Is the manuscript technically sound, and do the data support the conclusions?

Reviewer #1: No

Reviewer #2: Yes

2. Has the statistical analysis been performed appropriately and rigorously? 

Reviewer #1: N/A

Reviewer #2: Yes

3. Have the authors made all data underlying the findings in their manuscript fully available?

Reviewer #1: No

Reviewer #2: Yes

4. Is the manuscript presented in an intelligible fashion and written in standard English?

Reviewer #1: No

Reviewer #2: Yes

5. Review Comments to the Author

Reviewer #1: This is an important topic for two reasons: a once in a century pandemic, and a total ban on tobacco (and alcohol) represent opportunities to study the effects. Authors however have not provided a clear objective for the study. Can it be assumed that it is to assess how the media represented views on the pros and cons of a ban?

The methods are not well described. What is the universe of media considered? Seems only selected print media? What about information originating in or spread via social media (which is today the main source for information in South Africa with many studies showing low readership of traditional print media). Were efforts made to classify the stories in terms of their prominence in the publication (length, position, editorial as examples known this impacts on what is read). Was any data collected on readership by source in terms of numbers, demographics. geography? It is unclear why data were only collected from March 26 till August 17. Starting at least earlier in March and continuing for a few months would have provided a fuller sense of community and public health responses. For example, were there calls for cessation post the ban? Additional details about what was not banned are missing. Were NRTs easily available for example?

In the introduction, the fact that bans were placed on tobacco and alcohol is important context.

The results are difficult to interpret (see comments under methods). A few prominently placed articles which reach millions of people should be given greater weight then small stories in obscure publications. The analysis does not try to address this issue which is standard in analyses of media reach and impact. UCT School of Marketing or the Pretoria School of Business could have helped with this.

Conclusions vary between the abstract and the text. The main conclusion in the abstract being that media missed the opportunities to discuss smoking cessation during the pandemic (presumably they mean during the lockdown months as no data is presented for later periods.) This is not supported by any data provided nor is it discussed in the text of the conclusions where the focus is on industry interference and countering their statements.

The UCT School of Economics Tobacco Control program (funded by Gates, WHO, CRUK and others) completed several major reports during this period and after that paint a data based picture of what really happened: illicit trade soared, a new criminal class was established, billions of Rands were lost in from excise tax revenue; most smokers found ways to obtain cigarettes albeit counterfeit, illegal and illicit brands. Similar outcomes were reported by others re the alcohol ban. Further, social media erupted with stories of the above, advise re how to circumvent the bans, and concerns re revenue losses to small shopowners, spaza owners and others.

The authors are probably correct that more should have been done re promoting cessation (and harm reduction). UCT researchers suggest that the desire to cut consumption could have been achieved by a large increase in excise taxes supported by stronger controls on illicit trade.

Given the importance of this topic from a global policy perspective, research needs to be well designed and presented in ways that include all relevant issues. This paper does not do so,

Reviewer #2: This is a well written and presented manuscript. Its timely and raises an important topic that deserves more attention such as presented here. The paper could be strengthened by mentioning a bit more about harm reduction and other misinformation resulting in a further lost opportunity to speed up a move away from much more deadly combusted tobacco products. Moreover there is increasing research, carefully done, that suggests nicotine may actually be protective against covid-19 but smokers have also been misinformed about that .. see for example: https://www.qeios.com/read/UJR2AW.15 ... and there are newer recent studies reaffirming this finding.

Misinformation, especially errors of omission of important information also relates to the notion of harm reduced products and the missed opportunity to save lives due to the substantially less relative harms of non combusted nicotine delivery be it from Pharmacological nicotine in the form of Nicotine Replacement Therapy (NRT) as well as smokeless products such as e-cigarettes or smokeless tobacco relative to combusted toxic smoke inhalation but not nicotine. This is well covered in the Public Health England recent update, the gold standard Cochrane Collaborative reviews that e-cigarettes can help more smokers stop using combusted products than NRTs. and reach more smokers to save more lives.

Thus the main point or two to be added involves misleading press coverage to is detrimental to consumer decision making -- that is driven by reporters failure to check the strongest science (ie do their homework) before conveying misleading information to the public and especially by errors of omission regarding providing smokers with accurate information about the RELATIVE HARMS of toxic smoke from combustion compared with much less harms of nicotine without smoke as well as protective effects of nicotine on COVID-19 (or at least that the jury is out and its not clear smoking is directly causal in severity of Covid19 or even risk fo getting Covid19. again see recent articles that support the main conclusion of this review: https://www.qeios.com/read/UJR2AW.15

6. PLOS authors have the option to publish the peer review history of their article (what does this mean?). If published, this will include your full peer review and any attached files.

Reviewer #1: No

Reviewer #2: No

---

## [Author Response · Author response to Decision Letter 0]

7 Jul 2022

Response to reviewers’ feedback

PONE-D-22-09801

“Are they trying to control us people?”: News media coverage of COVID-19 lockdown tobacco sales ban in South Africa

PLOS ONE

Editor: Authors: We have done this. Thank you.

Journal Requirements:

Response: Done. Thank you.

Response: There was no external funding for this project. We have now deleted funding information from the manuscript

"COE and SPN’s time were supported by the South African Medical Research Council. EC and HB’s time was supported by the University of Nevada, Reno."

"COE and SPN’s time were supported by the South African Medical Research Council. EC and HB’s time was supported by the University of Nevada, Reno."

Response: Please change the funding statement to, “There was no external funding for this project.”

Reviewers' comments:

Comments to the Author

5. Review Comments to the Author

Reviewer #1: This is an important topic for two reasons: a once in a century pandemic, and a total ban on tobacco (and alcohol) represent opportunities to study the effects. Authors however have not provided a clear objective for the study. Can it be assumed that it is to assess how the media represented views on the pros and cons of a ban?

Response: Thank you. The objective has now been stated clearly to read “This study investigates news media coverage of arguments and activities in relation to the South African lockdown tobacco sales ban between March and August, 2020.” (page 6)

Comment: The methods are not well described. What is the universe of media considered? Seems only selected print media? What about information originating in or spread via social media (which is today the main source for information in South Africa with many studies showing low readership of traditional print media). 

Response: The universe of media has been included and the reasons for not including social media has been stated. See page 6: “We considered all media outlets with online presence (print, radio and TV) except social media (Facebook, Twitter, Instagram etc) which would have required a different methodology to be included.” We have also included the exclusion of social media as part of the limitations to the study (see page 24).

Also, please note that COVID-19 practically changed the landscape of South African media with online media readership experiencing an exponential growth especially as people searched for the latest COVID-19 news. This assertion has been documented in an article published in May 2022 here: https://mybroadband.co.za/news/internet/350502-how-covid-19-has-changed-south-africas-media-landscape.html

This publication also presents the readership of the first 20 news media outlet in South Africa and their year-on-year readership change.

Since our focus was on all media outlets with online presence, we believe we covered what most social media users also have access to. All of these media outlets we included in this study also have social media handles through which they share their news articles. Social media research if included in this study would duplicate news articles. We also believe a social media research would need a different methodology. We have included the below text to the limitations section of the paper.

Social media attracts huge readership in South Africa; however, most media outlets also maintain social media handles through which they further disseminate their news publications. Social media was not included in this study due to the difference in methodology and to avoid duplicating publications. For future research and practice it would be recommended to consider a distinct study on social media coverage and reactions to the COVID-19 lockdown tobacco ban and compare such results to our findings.

Were efforts made to classify the stories in terms of their prominence in the publication (length, position, editorial as examples known this impacts on what is read). 

Response: We made efforts to classify the publication based on the segment of the media where they were published. This is now included supplementary Table (Table S1).

Was any data collected on readership by source in terms of numbers, demographics. geography? 

Response: No, we could not get this information for all the articles due to the diversity of their sources hence it was not part of the methodology of assessing these publications. 

It is unclear why data were only collected from March 26 till August 17. Starting at least earlier in March and continuing for a few months would have provided a fuller sense of community and public health responses. For example, were there calls for cessation post the ban? 

Response: March 26 was when the ban commenced and August 17 was when the ban ended. It therefore made sense that we focused on this period because prior to the ban on tobacco products, there were no discussions about a ban as this happened rather quickly. Our paper is focused on capturing the discussions (arguments) and activities that were stimulated by the ban and not the entire lockdown period. Also, including more dates that were not directly relevant to the ban would unnecessarily widen the scope of the paper and would need the objective of the study to be redefined. 

Additional details about what was not banned are missing. Were NRTs easily available for example?

Response: There was no list of products which were banned but only categories of products which were not banned (https://www.thesouthafrican.com/lifestyle/list-of-essential-goods-during-lockdown-updated-friday-17-april-2020/). NRTs were not banned in South Africa during this period because they are classified and regulated as pharmaceutical products and not as tobacco or nicotine products hence the fall within the “medicine” category of items categorized as essential. We have included this information in the paper (page 5).

In the introduction, the fact that bans were placed on tobacco and alcohol is important context.

Response: Thank you. We have added this to the introduction (page 5).

The results are difficult to interpret (see comments under methods). 

Response: We have substantially revised the manuscript to enable easier understanding of the paper. Please, also see response to comments under methods.

A few prominently placed articles which reach millions of people should be given greater weight then small stories in obscure publications. The analysis does not try to address this issue which is standard in analyses of media reach and impact. UCT School of Marketing or the Pretoria School of Business could have helped with this.

Response: Thank you. We were qualitatively exploring the discussions and activities during this period as a way to appraise the issues of interest to the public during this period. However, we have noted and added your concerns as a limitation of the study (page 23).

Conclusions vary between the abstract and the text. The main conclusion in the abstract being that media missed the opportunities to discuss smoking cessation during the pandemic (presumably they mean during the lockdown months as no data is presented for later periods.) This is not supported by any data provided nor is it discussed in the text of the conclusions where the focus is on industry interference and countering their statements.

Response: Thank you for pointing this out. We have now made sure that the conclusion in the abstract and the text are in sync (see pages 2 and 24). See below.

Abstract conclusion: The media could serve as a veritable tool to promote public health if engaged in productive ways to communicate and promote public health regulations to the general population. Engagement with the media should be enhanced as part of health promotion strategies.

Paper conclusion: This study reveals the trends and themes of newspaper media coverage about the temporary ban on the sale of tobacco products in South Africa during lockdown. It shows the narrative the tobacco industry managed to push using the media and how they fought against the ban. At the same time, it also shows the activities and arguments of tobacco control advocates, and the government during the period of the tobacco ban. The media, being an important component of society, needs to be included in tobacco control efforts. Media platforms can be used to send the right information about government policies and programmes that would benefit tobacco control and public health in general. The media can play a huge role in spreading awareness or spreading misinformation as well as mobilizing the public in support for or against public health policies. The media may have been used to mobilize citizens against the ban. The health benefits of smoking cessation were scarcely discussed in the media during the period suggesting that this is a missed opportunity for public health. Industry monitoring and stricter laws which would prevent the tobacco industry from using the media to its advantage are needed while engagement with the media should be enhanced as part of health promotion strategies.

The UCT School of Economics Tobacco Control program (funded by Gates, WHO, CRUK and others) completed several major reports during this period and after that paint a data based picture of what really happened: illicit trade soared, a new criminal class was established, billions of Rands were lost in from excise tax revenue; most smokers found ways to obtain cigarettes albeit counterfeit, illegal and illicit brands. Similar outcomes were reported by others re the alcohol ban. Further, social media erupted with stories of the above, advise re how to circumvent the bans, and concerns re revenue losses to small shopowners, spaza owners and others.

Response: Thank you. However, the impact of the ban is not the focus of this paper. We qualitatively explored the discussions and activities in the media space during this period as a way to appraise the issues of interest to the public during this period. The claims about illicit trade and loss in tax revenue were included in the results (see pages 19 and 21). These claims were also included as under anti-tobacco ban activities and arguments on theme 5, subtheme 4. We also did not include social media as part of our media universe for reasons now added to the limitation of the study. 

The authors are probably correct that more should have been done re promoting cessation (and harm reduction). UCT researchers suggest that the desire to cut consumption could have been achieved by a large increase in excise taxes supported by stronger controls on illicit trade.

Response: Thank you. We do agree with colleagues at UCT about alternative approaches to achieving the aim of the ban. However, as mentioned earlier, our paper is not focused on evaluating the approach by government to curb tobacco consumption. 

Given the importance of this topic from a global policy perspective, research needs to be well designed and presented in ways that include all relevant issues. This paper does not do so,

Response: We have used the reviewer’s comments to clarify the paper. Thank you. 

Reviewer #2: This is a well written and presented manuscript. Its timely and raises an important topic that deserves more attention such as presented here. 

Response: Thank you very much.

The paper could be strengthened by mentioning a bit more about harm reduction and other misinformation resulting in a further lost opportunity to speed up a move away from much more deadly combusted tobacco products. Moreover there is increasing research, carefully done, that suggests nicotine may actually be protective against covid-19 but smokers have also been misinformed about that .. see for example: https://www.qeios.com/read/UJR2AW.15 ... and there are newer recent studies reaffirming this finding.

Response: The question of the relationship between smoking and other forms of tobacco use remains controversial. While the preprint the reviewer cites supports the idea that smoking may be protective against COVID19 infection, other data contradict that conclusion. Other studies do not support the hypothesis that nicotine could be protective against COVID19. For example, see: Lallai, V., Manca, L., & Fowler, C. D. (2021). E-cigarette vape and lung ACE2 expression: Implications for coronavirus vulnerability. Environmental Toxicology and Pharmacology, 86, 103656. doi:10.1016/j.etap.2021.103656. Addressing this major issue is a different question than addressed by the data we present in our paper.

Misinformation, especially errors of omission of important information also relates to the notion of harm reduced products and the missed opportunity to save lives due to the substantially less relative harms of non combusted nicotine delivery be it from Pharmacological nicotine in the form of Nicotine Replacement Therapy (NRT) as well as smokeless products such as e-cigarettes or smokeless tobacco relative to combusted toxic smoke inhalation but not nicotine. This is well covered in the Public Health England recent update, the gold standard Cochrane Collaborative reviews that e-cigarettes can help more smokers stop using combusted products than NRTs. and reach more smokers to save more lives.

Response: Like the COVID19 and nicotine question, e-cigarettes’ value for cessation is beyond the scope of this paper. While the Cochrane reviews and other meta-analyses of RCTs of e-cigarettes as supervised medicine do show efficacy as a form of NRT, the population studies of unsupervised use of e-cigarettes as consumer products show no cessation benefit Kindly note that NRTs were not banned in South Africa during this period because they are classified and regulated as pharmaceutical products and not as tobacco or nicotine products. Please see Richard J. Wang, Sudhamayi Bhadriraju, Stanton A. Glantz, “E-Cigarette Use and Adult Cigarette Smoking Cessation: A Meta-Analysis”, American Journal of Public Health 111, no. 2 (February 1, 2021): pp. 230-246. https://doi.org/10.2105/AJPH.2020.305999 s well as a South African study: Agaku, I., Egbe C.O., & Ayo-Yusuf, O.A. (2021). Associations between electronic cigarette use and quitting behaviors among South African adult smokers. Tobacco Control. doi: https://doi.org/10.1136/tobaccocontrol-2020-056102.

Also, NRT and other therapeutic forms of nicotine were not banned. This point has been added to the paper.

Thus the main point or two to be added involves misleading press coverage to is detrimental to consumer decision making -- that is driven by reporters failure to check the strongest science (ie do their homework) before conveying misleading information to the public and especially by errors of omission regarding providing smokers with accurate information about the RELATIVE HARMS of toxic smoke from combustion compared with much less harms of nicotine without smoke as well as protective effects of nicotine on COVID-19 (or at least that the jury is out and its not clear smoking is directly causal in severity of Covid19 or even risk fo getting Covid19. again see recent articles that support the main conclusion of this review: https://www.qeios.com/read/UJR2AW.15

Response: Please see previous responses on these issues raised. Thank you.

---

## [Decision Letter · Decision Letter 1]

25 Aug 2022

PONE-D-22-09801R1“Are they trying to control us people?”: News media coverage of COVID-19 lockdown tobacco sales ban in South AfricaPLOS ONE

Dear Dr. Egbe,

Thank you for submitting your manuscript to PLOS ONE. After careful consideration, we feel that it has merit but does not fully meet PLOS ONE’s publication criteria as it currently stands. Therefore, we invite you to submit a revised version of the manuscript that addresses the points raised during the review process.

Your manuscript has been reassessed by one reviewer from the previous round, whose reports can be found below. Whilst the manuscript has improved significantly, there remains a yet to resolved issue about the harm reduction continuum which should be addressed before your manuscript is suitable for publication.

A marked-up copy of your manuscript that highlights changes made to the original version. You should upload this as a separate file labeled 'Revised Manuscript with Track Changes'.An unmarked version of your revised paper without tracked changes. You should upload this as a separate file labeled 'Manuscript'.

We look forward to receiving your revised manuscript.

Kind regards,

Katrien Janin

Staff Editor

PLOS ONE

Journal Requirements:

Reviewers' comments:

Reviewer's Responses to Questions

**Comments to the Author**

1. If the authors have adequately addressed your comments raised in a previous round of review and you feel that this manuscript is now acceptable for publication, you may indicate that here to bypass the “Comments to the Author” section, enter your conflict of interest statement in the “Confidential to Editor” section, and submit your "Accept" recommendation.

Reviewer #2: (No Response)

2. Is the manuscript technically sound, and do the data support the conclusions?

Reviewer #2: Yes

3. Has the statistical analysis been performed appropriately and rigorously? 

Reviewer #2: Yes

4. Have the authors made all data underlying the findings in their manuscript fully available?

Reviewer #2: Yes

5. Is the manuscript presented in an intelligible fashion and written in standard English?

Reviewer #2: No

6. Review Comments to the Author

Reviewer #2: I do not feel the most important comment about the harm reduction continuum and use of much less harmful non combusted consumer products (not just NRT) was addressed in the revision. at the very least a coup[le sentences in support of harm reduction and stronger citing of balanced literature rather than the selective use of very weak science and misleading statements by Dr Standton Glants and co authors are NOT a good reason to exclude some framing of the potential for non combusted nictine delivery - along the harm reduction continuum --to be used as a way to make combusted smoking pbsolete more rapidly and pragmatically. This is not beyond the scope of this paper at all and should be included. Moreover the authors seem to themselves adhere to the misinformation promulgated by the weakest science against e-cigarettes (vaped nicotine) and selectively ignore the much stronger scientific evidence in support of harm reduction as a pragmatic solution to the obvious failure of different attempt at total nicotin e prohibition (an extremist view that may be both unrealistic and unreachable -- and totally relevant to the main focus of this paper.

The UK approach and Public Health England's excellent reviews and updates as well as stronger balances science from the USA cannot simply be dismissed as beyond the scope of this paper -- indeed that is the main GOAL of the public health approach at the population level -- elimination of the largely preventable deaths and massive chonic disease burdens of decades of COMBUSTED SMOKE INHALATION and toxins in burned tobacco and not of nicotine when delivered in non-combusted but appealing forms other than medicinal NRT where there is clear evidence of much lower levels and much fewer types of toxins in e-cigarette vapor , and in use of Swedish type snus and newer nicotine pouches and so forth. There is ample evidence of dramatic reductions in biomarkers of harm especially major cancer bio-makers such as from FDAs own PATH analysis of toxins in e-cigarette users blood versus combusted users . the major USA Institute of Medicine, the FDAs own 2017 strategy for managing nicotine (Gottlieb and Zeller ); several well done independent reviews ; many of the best studies of exposure and many critical comments about the limitations of poorly done studies that are often used to mislead. Including one study by Glantz et al -- on e-cigarettes and heart attacks --that had to be and was withdrawn by the journal (Am Heart Assn) and others that should have been withdrawn.

7. PLOS authors have the option to publish the peer review history of their article (what does this mean?). If published, this will include your full peer review and any attached files.

Reviewer #2: No

---

## [Author Response · Author response to Decision Letter 1]

26 Aug 2022

Please check our document titled: response to the reviewiers

---

## [Decision Letter · Decision Letter 2]

5 Oct 2022

PONE-D-22-09801R2“Are they trying to control us people?”: News media coverage of COVID-19 lockdown tobacco sales ban in South AfricaPLOS ONE

Dear Dr. Egbe,

Thank you for submitting your manuscript to PLOS ONE. After careful consideration, we feel that it has merit but does not fully meet PLOS ONE’s publication criteria as it currently stands. Therefore, we invite you to submit a revised version of the manuscript that addresses the points raised during the review process.

We look forward to receiving your revised manuscript.

Kind regards,

Ali B. Mahmoud, Ph.D.

Academic Editor

PLOS ONE

Journal Requirements:

Reviewers' comments:

Reviewer's Responses to Questions

**Comments to the Author**

1. If the authors have adequately addressed your comments raised in a previous round of review and you feel that this manuscript is now acceptable for publication, you may indicate that here to bypass the “Comments to the Author” section, enter your conflict of interest statement in the “Confidential to Editor” section, and submit your "Accept" recommendation.

Reviewer #2: (No Response)

2. Is the manuscript technically sound, and do the data support the conclusions?

Reviewer #2: Yes

3. Has the statistical analysis been performed appropriately and rigorously? 

Reviewer #2: N/A

4. Have the authors made all data underlying the findings in their manuscript fully available?

Reviewer #2: Yes

5. Is the manuscript presented in an intelligible fashion and written in standard English?

Reviewer #2: Yes

6. Review Comments to the Author

Reviewer #2: While the added sentence is somewhat responsive about the harm continuum and electronic or non combusted nicotine products -- the authors continue to state that all the concern about exempting electronic cigarettes is industry misinformation aboyt harm reduction . Thus it does not adequately address the issue that scientific opinion also provides scientific support for harm reduction and the substitution of much less harmful nicotine products cigarettes can save lives. The sentence could be acceptable - although its minimal - if several credible references in support of the science to date were cited , perhaps the least controversial of these are the following two citations (Public Health England latest scientific update and the very conservative review by 15 presidents of the worlds leading research society on nicotine and tobacco (SRNT). (The issue of Glantz and a personal vendetta is a distraction from the authors addressing adequately the real issue in their discussion and their own perhaps unintended bias in their interpretations of their own results -- as all evil industry related ). The issue is to take the highest ground of scientific evidence about harm reduced products and call for media accuracy based on science not industry past behavior ( and to indicate that despite the obvious and despicable industry tactics in general - which I agree with -- in terms of the industry being opposed to the entire ban etc, BUT as the authors stated -- one needs more press coverage with accurate science as well to counter industry tactics that do not protect public health but profits and their more massive budgets as the authors state. BUT again there IS ALSO scientific evidence (NOT ALL Industry misinformation) about the value and evidence in support of exempting electronic cigarettes / non combustible nicotine -- and correct misinformation in the media about a harm continuum of products -- so as to accelerate e and informed consumer decision making and government policy -- to save more smokers lives more quickly . So that can be clarified a bit mor strongly in the added sentence AND alos please ADD the two references below. These are from most credible sources and a full review of all the evidence to date . They qualify the weak and still therefore misleading added sentence from the authors -- one that suggests there are equal pro and con arguments for and against e-cigarettes (not a continuum of harm) even from a science point of view. This is not true ( i.e. thus by implication e-cigs are not less harmful and that any harm reduction claim is still to be ignored as its all part of an industry misinformation campaign. Harm reduction is not ONLY an industry plot, it is more and more strongly scientifically supported. I agree there was not enough in depth coverage of e-cigarettes as stated in the revision , but it does not clarify the core issue to ensure accurate media coverage of the harm continuum and reduced harm products to save lives. The authors can do better at composing a few sentences than their current added sentence.

1. Balancing Consideration of the Risks

and Benefits of E-Cigarettes. David J. K. Balfour, DSc, Neal L. Benowitz, MD, Suzanne M. Colby, PhD, Dorothy K. Hatsukami, PhD, Harry A. Lando, PhD,

Scott J. Leischow, PhD, Caryn Lerman, PhD, Robin J. Mermelstein, PhD, Raymond Niaura, PhD, Kenneth A. Perkins, PhD, Ovide F. Pomerleau, PhD, Nancy A. Rigotti, MD, Gary E. Swan, PhD, Kenneth E. Warner, PhD, and Robert West, PhD.

Am J Public Health. Published online ahead of print

August 2021:e1–e12. https://doi.org/10.2105/AJPH.2021.306416)

2. McNeill A, Brose LS, Calder R, Simonavicius E,

Robson D. Vaping in England: An Evidence Update

Including Vaping for Smoking Cessation, February

2021. A Report Commissioned by Public Health

England. Public Health England. 2021. Available

at: https://assets.publishing.service.gov.uk/

government/uploads/system/uploads/

attachment_data/file/962221/Vaping_in_

England_evidence_update_February_2021.pdf.

Accessed March 6, 2021.

7. PLOS authors have the option to publish the peer review history of their article (what does this mean?). If published, this will include your full peer review and any attached files.

Reviewer #2: No

---

## [Author Response · Author response to Decision Letter 2]

26 Oct 2022

Please see response to the reviewer document

---

## [Editor Report · Decision Letter 3]

28 Nov 2022

“Are they trying to control us people?”: News media coverage of COVID-19 lockdown tobacco sales ban in South Africa

PONE-D-22-09801R3

Dear Dr. Egbe,

We’re pleased to inform you that your manuscript has been judged scientifically suitable for publication and will be formally accepted for publication once it meets all outstanding technical requirements.

Kind regards,

Al B. Mamod, Ph.D.

Academic Editor

PLOS ONE
---

## [Editor Report · Acceptance letter]

2 Dec 2022

PONE-D-22-09801R3 

“Are they trying to control us people?”: News media coverage of COVID-19 lockdown tobacco sales ban in South Africa 

Dear Dr. Egbe:

I'm pleased to inform you that your manuscript has been deemed suitable for publication in PLOS ONE. Congratulations! Your manuscript is now with our production department. 

Kind regards, 

on behalf of

Dr. Ali B. Mahmoud 

Academic Editor

PLOS ONE